# Perinatal and early life risk factors of adverse early childhood developmental outcomes: Protocol for systematic review using socioecological model

Kendalem Asmare Atalell [1,2]*, Gavin Pereira[2,3], Bereket Duko[2,4,5], Sylvester Dodzi Nyadanu[2], Gizachew A. Tessema[2,3,6]

1 College of Medicine and Health Sciences, University of Gondar, Gondar, Ethiopia, 2 Curtin School of Population Health, Curtin University, Perth, Western Australia, Australia, 3 enAble Institute, Curtin University, Bentley, WA, Australia, 4 Australian Centre for Precision Health, University of South Australia, Adelaide, South Australia, Australia, 5 South Australian Health and Medical Research Institute, Adelaide, South Australia, Australia, 6 School of Public Health, University of Adelaide, Adelaide, South Australia, Australia

* k.atalell@postgrad.curtin.edu.au

**Data Availability Statement:** No datasets were generated or analysed during the current study. All relevant data are included in the manuscript.

## Abstract

### Background

Adverse early childhood developmental outcomes impact later schooling and adulthood life courses. However, there needs to be more comprehensive evidence on the effect of various perinatal and early life risk exposures. Hence, we aimed to systematically identify the various perinatal and early childhood risk factors using a socioecological model to inform appropriate prevention strategies.

### Method

The systematic review will adhere to the 2020 PRISMA guidelines. The protocol was registered in PROSPERO with a registration number of CRD42023447352. We will systematically search for articles on adverse early childhood developmental outcomes, which include physical, cognitive, language and communication and social-emotional development from main databases, such as EMBASE, Medline, Global Health, PsycINFO, CINAHL and Web of Science Core Collection, dating from 2000. We will use Population, Exposure, Comparator, Outcome and Study Design (PECOS) criteria to select eligible studies for our review: 1) Population includes children in early childhood age (i.e., up to eight years) undergoing developmental assessments. 2) Exposure: various perinatal and early life risk factors. 3) Comparators: Children with no or low levels of exposure to the risk factors. 4) Outcome: adverse early childhood developmental outcome. 5) Study design: all observational studies that report the prevalence or incidence of adverse early childhood developmental outcomes and associated risk factors published since 2000. There will be no restriction based on country of origin or geographical location except language (only published in English). Textual and narrative synthesis using the socioecological model will be used to synthesise the data.

**Funding:** The author(s) received no specific funding for this work.

**Competing interests:** The authors declared that there is no conflict of interest to declare.

## Introduction

Early childhood, as defined by the World Health Organization (WHO), spans from prenatal development to eight years of age. It serves as a critical foundation for the entire life course [1, 2]. This period is crucial due to the rapid growth and development, encompassing physical, cognitive, language and communication and social-emotional developmental domains, which are closely linked with basic learning, school success, economic participation, social citizenry, and health outcomes [3–6]. Adversities faced during early childhood periods can lead to long-lasting effects such as mental health issues, poor literacy, unemployment, criminality, violence, and poor health outcomes in later life [3, 7]. The early childhood period is also sensitive to potential risk factors, and timely interventions can promote healthy development, academic success, and overall well-being [8]. Adverse early child developmental outcomes, which refer to difficulties or delays in various areas of child development, including physical, cognitive, language communication, and social-emotional domains, were the major challenges in early childhood [9, 10]. Diverse assessment tools such as the Early Development Instrument (EDI), [11, 12] the Ages and Stages Questionnaires (ASQ) [13–15], the Bayley Scales of Infant and Toddler Development (Bayley) [16] and the Denver Developmental Screening Test (DDST) [17] were among the commonly used tools to evaluate early childhood developmental adversities. Understanding the various domains of early childhood developmental outcomes assessed through various developmental assessment tools and exploring the various risk factors associated with early childhood developmental adversities would inform policymakers on the prevention and health promotion activities to better support early childhood development.

At the age of five years, an estimated 250 million children did not achieve their full developmental potential, mostly in low and middle-income countries, representing 43% of the under-five children in 2016 [8]. Previous studies have demonstrated that early childhood developmental outcomes were affected by various perinatal and early life risk exposures. These include genetic and biological factors, prenatal conditions such as maternal health, nutrition, exposure to toxic substances, prenatal care during pregnancy, birth complications, child health, parenting behaviours, access to early childhood education and care, socioeconomic disadvantages, and environmental exposure to toxins or pollutants [9, 18–25]. Comprehensively summarizing the risk factors of adverse early childhood developmental outcomes might help to develop strategies to mitigate developmental adversities.

Even though efforts have been made to combat early childhood developmental adversities through global initiatives like the Sustainable Development Goals (SDGs), specifically target 4.2, which aims to ensure that all girls and boys have access to quality early childhood development, care, and pre-primary education by 2030, many children, particularly in low and middle-income countries, still lack access to these crucial developmental resources [26–31]. Previous systematic reviews have assessed risk factors influencing early childhood developmental outcomes [25, 32–35] but often focused on single exposures or specific domains of adverse developmental outcomes, limiting comprehensive syntheses of multilevel risk factors [36]. On the other hand, the current study aims to summarize risk factors affecting adverse early childhood developmental outcomes across individual, interpersonal, community and societal levels, as suggested by the socioecological model. The socio-ecological model is a theoretical framework used to understand and analyze the complex interaction between 1) Individual factors: factors arise from the individual child such as genetics, health status, lifestyle and behavioral factors. 2) Interpersonal factors: maternal, paternal, or household-related influences, including family dynamics and parenting styles. 3) Community factors: school, peers, teachers, daycare environments, and neighborhood-related influences. 4) Societal factors: broader influences include climatic conditions, cultural norms, systems, policies, and

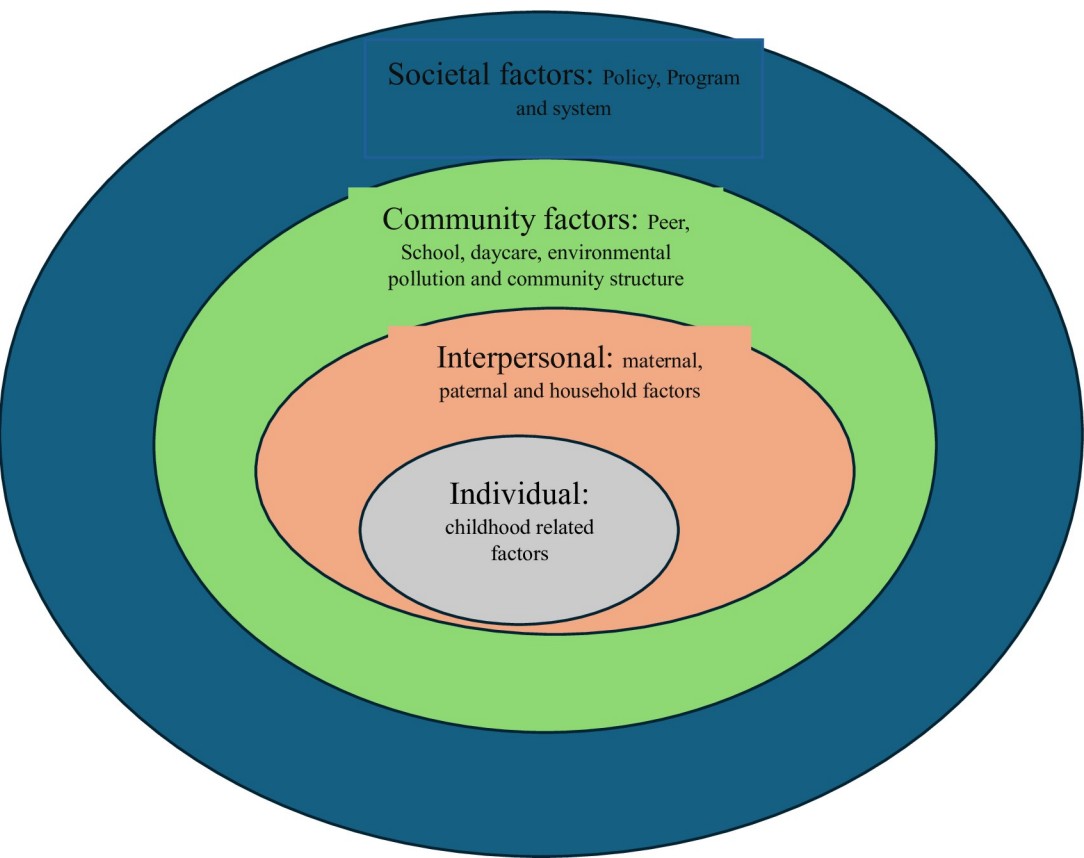

**Fig 1. Socioecological model framework used to summarise the various perinatal and early life risk exposures towards adverse early childhood developmental outcomes.**

programs Fig 1. The socio-ecological model is crucial in recommending a targeted intervention at each level. Hence, identifying the multilevel associated risk factors is imperative for identifying vulnerable populations and implementing targeted health promotion activities at each level [8].

Therefore, this systematic review aimed to comprehensively map the broader risk factors of early childhood developmental adversities.

## Research questions

This systematic review is intended to answer the following research questions:

1. What are the perinatal and early childhood risk factors for adverse early childhood developmental outcomes?

## Method

This systematic review and meta-analysis will adhere to the Preferred Reporting Items for Systematic Reviews and Meta-Analyses (PRISMA) guidelines 2020 (S1 Table) [37], and the protocol has been registered with PROSPERO under registration number CRD42023447352 [38]. Additionally, we will follow the Joanna Briggs Institutes (JBI) manual for evidence synthesis [39, 40] to ensure comprehensive reporting of our systematic review results.

## Eligibility criteria

We used the Population, Exposure, Comparison, Outcome and Studies (PECOS) criteria to select eligible studies for our study in the following format.

1. **Population:** Children under eight years old underwent early childhood developmental assessments. The age of 8 years is chosen because early childhood is defined as the period from prenatal development to eight years, according to WHO.

2. **Exposure:** The exposures in this study include various perinatal and early childhood risk factors, such as biological, sociodemographic, child health and medical history, childhood nutrition, maternal mental health issues, pregnancy-related maternal health conditions, maternal nutrition, prenatal exposure to substances and pollutants, environmental pollutants, and climatic factors.

3. **Comparison:** Children with no or low levels of exposure to the risk factors will be used as a comparator for this study. Low level of exposure to the risk factors refers to individuals or groups who have minimal exposure to the identified risk factors compared to the general population, this will be determined based on specific thresholds or criteria established in the included studies.

4. **Outcomes:** The primary outcome of this systematic review is adverse early childhood developmental outcomes, such as physical, cognitive, language and communication and social-emotional developments [41, 42], which is defined as difficulties or delays in various areas of child developmental vulnerability or developmental problems at composite or sub-domain levels. These outcomes are assessed through various childhood developmental assessment tools such as the Early Development Instrument (EDI), Age and Stages Questionnaire (ASQ), Bayley Scales of Infant and Toddler Development (Bayley), Denver Developmental Screening Test (DDST), etc. [41, 42].

5. **Studies:** We will include observational studies such as cohort, case-control, nested case-control, and cross-sectional studies reporting on risk factors associated with developmental outcomes in children up to eight years of age according to the WHO definition [43]. There will be no restriction based on country of origin or geographical location for inclusion, except must be published in English from January 2000 to January 2024. The year 2000 was chosen to capture the advancements in technology, changes in social media use, social and cultural contexts, developments in early childhood education, and shifts in environmental exposures. This approach aims to incorporate recent and up-to-date knowledge in the field, reflecting significant changes in policies, interventions, and programs over the past two decades to improve early childhood developmental outcomes.

## Exclusion criteria

We will exclude studies such as commentaries, letters to editors, conference proceedings, case reports, case series, correspondence, interventional studies, systematic reviews and meta-analyses and abstracts.

## Information sources and search strategy

We will systematically search for articles on adverse early childhood developmental outcomes across several databases, including EMBASE, MEDLINE, Global Health, PsycINFO, CINAHL, SCOPUS, and Web of Science Core Collection from January 2000 to January 2024. Additionally, we will conduct forward and backward reference searching to identify relevant articles

and search Google Scholar. The search strategy will be conducted in three stages following JBI guidelines: i) we will develop keywords and their synonyms combined with Boolean operators in three key concepts, namely population (children), outcome (developmental outcomes) and exposures (prenatal, early life, biological, psychosocial, behavioural, and environmental risk factors); ii) Appropriate Medical Subject Heading (MeSH) terms will be identified and searched in each database by combining with keywords in each key concept; and iii) we will search google scholar and Google for grey literature (S2 Table). The first 50 hits will be reviewed for relevant literature in Google Scholar and Google. Backward and forward searching of the included studies will also be performed. The specific search terms for each database will be available upon request and published with the result of the systematic review prospectively.

## Study selection process

The articles searched in each database will be imported into the EndNote 20 reference manager software. After removing duplicates in EndNote, we will import the studies to Rayyan, an online systematic review program used to facilitate the screening of abstracts and full text. Studies with relevant titles and abstracts will undergo a full-text review based on the predefined inclusion criteria. Two authors will conduct the screening process; a third reviewer will resolve any conflicting issues through discussion. Ineligible studies will be excluded with reasons and presented with a PRISMA flow diagram.

## Data extraction

Data extraction format will be prepared in a Microsoft Excel 2019 and piloted in 10 hand-searched articles before use. The data extraction format will capture data from the included studies, including the first author's name and country study originated, the aim of the study, study design, publication year, study population, sample size, WHO region, mean or median age, the age at which child development assessment was taken, measurement tools used to assess the developmental outcomes, perinatal and early life risk factors measured by effect measures such as odds ratios (OR), relative risks (RR), or β coefficient and the primary outcome. If effect estimates are not reported in the included studies, we will extract the following data: the number of exposed individuals, the number of non-exposed individuals, the number of cases among the exposed group, and the number of cases among the non-exposed group (S3 Table). Two authors will extract the data on each included study. Given the anticipated large volume of studies in our systematic review, we plan to conduct double data extraction for approximately 20% of the included studies. After ensuring consistency in the extracted data, we will proceed to extract the remaining data, with each author handling 50% of the remaining studies.

## Quality or risk of bias assessment

The methodological quality of each included study will be assessed using the JBI Critical Appraisal checklist for different study designs. The JBI critical appraisal checklists contain yes/ no or unclear responses and are sometimes not applicable [44]. To assess the overall risk of bias in each study, we will assign a score (yes = 1) if the paper met the criteria or (no/ unclear = 0) if the paper doesn't meet the criteria of unclear. After summing up the scores, a paper with a high score will be considered high-quality, and those with low scores will be considered lower quality as applied elsewhere.

### Data synthesis

Textual and narrative syntheses of the early childhood developmental outcomes will be conducted in each developmental domain, such as physical, cognitive, language and communication, social-emotional and assessment tools. Then, we will thematize the developmental outcomes in each subdomain and levels of risk factors using the socioecological model, will be used to map the complex interplay between various biological, psychosocial, behavioural, perinatal and environmental factors associated with adverse early childhood developmental outcomes, which helps to target health promotion activities at each level [45, 46]. Based on the preliminary literature review [47–54], the four-level socioecological models will be formed. 1) Individual level factors: This level includes characteristics that arise from the child, such as age, sex, nutrition status, comorbid conditions, etc. 2) Interpersonal or household-level factors: includes maternal, paternal, and household-level factors or exposures. 3) Community-level factors include risk factors related to peers, schools, preschools, neighbourhoods, early childhood education and care centres, daycare, community customs, environmental exposures, and climatic factors. 4) Societal-level factors encompass national-level policy, program, and system-related risk factors (Fig 1).

### Ethics statements

No ethics approval is needed for this project as we will rely on publications available in the evidence base.

## Discussion

Every child has the right to access good early childhood developmental care [55]. Scientific evidence supports the notion that intervening as early as possible is crucial for promoting, protecting, safeguarding, and nurturing a child's development. This is particularly vital during pregnancy and the first five years of life, representing a critical phase in human development that significantly impacts their well-being, cognitive abilities, and long-term health [8, 28]. Collaboration among healthcare providers, educators, policymakers, families, and communities is essential to ensure access to prenatal care, nutrition, early education, and supportive environments [27]. Global initiatives like the Global Strategy for Women's, Children's and Adolescent Health and SDG [26] have been established to improve early childhood developmental outcomes. This systematic review aims to understand global early childhood developmental adversities and identify risk factors and their interplay. The findings will benefit families, healthcare providers, policymakers, and researchers through generating evidence-based health promotion recommendations. One of the anticipated limitations of our systematic review is the heterogeneity in early childhood developmental assessment tools, which could lead to variability in how developmental outcomes and their domains are reported. This diversity might introduce challenges in synthesizing the data and drawing consistent conclusions across studies. To address this, we will employ the socioecological model, which provides a comprehensive framework to summarize and interpret the relationships between various risk factors and early developmental outcomes.

## Supporting information

**S1 Table. PRISMA-P (Preferred Reporting Items for Systematic Review and Meta-Analysis Protocols) 2015 checklist: Recommended items to address in a systematic review protocol.** (DOC)

**S2 Table. Search strategy.**
(DOCX)

**S3 Table. Data extraction checklist.**
(XLSX)

## Author Contributions

**Conceptualization:** Kendalem Asmare Atalell, Gavin Pereira, Bereket Duko, Sylvester Dodzi Nyadanu, Gizachew A. Tessema.

**Data curation:** Kendalem Asmare Atalell, Gizachew A. Tessema.

**Investigation:** Kendalem Asmare Atalell.

**Methodology:** Kendalem Asmare Atalell, Gizachew A. Tessema.

**Project administration:** Kendalem Asmare Atalell, Gizachew A. Tessema.

**Supervision:** Gavin Pereira, Bereket Duko, Sylvester Dodzi Nyadanu, Gizachew A. Tessema.

**Validation:** Gavin Pereira, Bereket Duko, Sylvester Dodzi Nyadanu, Gizachew A. Tessema.

**Writing – original draft:** Kendalem Asmare Atalell.

**Writing – review & editing:** Kendalem Asmare Atalell, Gavin Pereira, Bereket Duko, Sylvester Dodzi Nyadanu, Gizachew A. Tessema.

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
