## [Decision Letter · Decision Letter 0]

20 Feb 2024

PONE-D-23-35644Perinatal and early life risk factors of early childhood developmental vulnerability: Protocol for systematic review and meta-analysis.PLOS ONE

Dear Dr. Atalell,

Thank you for submitting your manuscript to PLOS ONE. After careful consideration, we feel that it has merit but does not fully meet PLOS ONE’s publication criteria as it currently stands. Therefore, we invite you to submit a revised version of the manuscript that addresses the points raised during the review process.

Specifically, the reviewer requests clarification around the methodology, such as provision of an explanation (and citation) for the socioecological model, additional background information, and suggests reconsidering the suggested subgrouping which may be too broad. Please see the specific comments below. 

We look forward to receiving your revised manuscript.

Kind regards,

Jennifer Tucker, PhD

Associate Editor

PLOS ONE

Journal Requirements:

Additional Editor Comments:

Please note that we have only been able to secure a single reviewer to assess your manuscript. We are issuing a decision on your manuscript at this point to prevent further delays in the evaluation of your manuscript. Please be aware that the editor who handles your revised manuscript might find it necessary to invite additional reviewers to assess this work once the revised manuscript is submitted. However, we will aim to proceed on the basis of this single review if possible. 

Reviewers' comments:

Reviewer's Responses to Questions

**Comments to the Author**

1. Does the manuscript provide a valid rationale for the proposed study, with clearly identified and justified research questions?

Reviewer #1: Partly

2. Is the protocol technically sound and planned in a manner that will lead to a meaningful outcome and allow testing the stated hypotheses?

Reviewer #1: Partly

3. Is the methodology feasible and described in sufficient detail to allow the work to be replicable?

Reviewer #1: No

4. Have the authors described where all data underlying the findings will be made available when the study is complete?

Reviewer #1: No

5. Is the manuscript presented in an intelligible fashion and written in standard English?

Reviewer #1: Yes

6. Review Comments to the Author

You may also provide optional suggestions and comments to authors that they might find helpful in planning their study.

Reviewer #1: This is a fair protocol but some details are missing. Adding that clarity now will make the review process easier. There are many studies on this topic so a really comprehensive systematic review, done well, is a helpful contribution to the field. However there are similar entries on PROSPERO, see here:https://www.crd.york.ac.uk/prospero/display_record.php?RecordID=3215

Please ensure your contribution is a novel contribution to the field. Your registered PROTOCOL uses the word 'adversities' rather than 'vulnerabilities'. The terms need to be consistent.

Introduction

The concept of early childhood development vulnerability needs to be explained early on. Please define this and if it has been derived from other sources please cite accordingly.

Other studies that have looked at the impact of development at school entry and trajectories of development on adolescent health and may be worth a look if you are making a life course argument https://www.mdpi.com/1660-4601/18/21/11613

and https://www.jpeds.com/article/S0022-3476(23)00474-2/fulltext and want to cite the most recent research.

Classification system for child development is also discussed here: https://www.mdpi.com/1660-4601/18/21/11613

Please provide a simple explanation (and citation) for the socioecological model.

The existing evidence on what is known in this field is more comprehensive that the lines portrayed: 'There were some

fragmented systematic syntheses of evidence such as child developmental outcomes associated

with low birth weight and small for gestational age in Indigenous Children from Australia,

Canada and New Zealand.19 '

Please provide a brief but comprehensive summary on what is known about risk factors (state which ones) that affect ECD vulnerability and state which countries. (This is why your definition of ECD vulnerability is needed right from the outset).

Research Qs - is research question 2 about 'predictive' risk factors?

Eligibility:

This section needs to be much clearer. The population blurs with the Concept. Suggest: Population (children 0-5), Exposure (risk factors), Outcomes (developmental) and Study types. See here: https://systematicreviewsjournal.biomedcentral.com/articles/10.1186/s13643-021-01694-6

The risk factors proposed here need to be introduced more clearly in the introduction.

Data Synthesis: The terms 'domains' and measurement tools are suddenly introduced. Please be consistent in language and terms throughout the document. It states you expect a variety of 'early child developmental outcome domains' - why has the language changed from ECD vulnerabilities to developmental outcomes? Your search strategy only lists negative developmental outcomes. Again - this must all be clarified in the eligibility section and then all the terms must remain consistent. E.g. - is the 'outcome' vulnerability (sub-optimal development) or any developmental domain?

It appears there is narrative and quantitative synthesis. Please describe both clearly including a section on confidence in cumulative evidence (i.e. for any narrative synthesis - how will you determine the strength of the evidence?)

The subgroup analysis is too broad to be well thought out. Perhaps think about what subgroup analysis would be useful?

Can Quality assessment templates and data extraction templates be included as supplementary files?

7. PLOS authors have the option to publish the peer review history of their article (what does this mean?). If published, this will include your full peer review and any attached files.

Reviewer #1: No

---

## [Author Response · Author response to Decision Letter 0]

26 Mar 2024

Ref: PONE-D-23-35644

Dr Jennifer Tucker,

Editor, PLOS ONE 

Title: Perinatal and early life risk factors of early childhood developmental vulnerability: Protocol for systematic review and meta-analysis.

Authors: Kendalem Asmare Atalell, Gavin Pereira, Bereket Duko, Sylvester Dodzi Nyadanu, Gizachew A Tessema

Subject: Submission of revised manuscript

Dear Editor,

Thank you for the opportunity to revise our manuscript “Perinatal and Early Life Risk Factors of Early Childhood Developmental Vulnerability: Protocol for Systematic Review and Meta-analysis”. We thank the reviewers and editors for their constructive and insightful comments and suggestions. We have now revised the manuscript and incorporated the comments. Our responses are given in point-by-point responses to each reviewer and the editor’s comments below. We have also attached the revised manuscript with tracked changes and a clean version.

We hope the revised version is suitable for publication and look forward to hearing from you soon.

Sincerely

Kendalem Asmare Atalell

Corresponding Author

Response to the editor and reviewers’ comments,

Response to the editor 

Comment #1: Specifically, the reviewer requests clarification around the methodology, such as the provision of an explanation (and citation) for the socioecological model and additional background information, and suggests reconsidering the suggested subgrouping, which may be too broad. 

Response: As requested by reviewers, we have now revised the manuscript to clarify the methodology and provide additional explanation for the socio-ecological model on page 8 lines 219-232 on the revised manuscript. “The socio-ecological model is important to map the complex interplay between various biological, psychological, behavioural, perinatal and environmental risk factors of adverse early childhood developmental outcomes into multilevel factors that include intrapersonal, interpersonal, community and societal levels, 50 that help to target health promotion activities at the individual, family, community or societal level. The four-level socioecological model will be formed based on the preliminary literature review. The first level is the intrapersonal risk factors or individual/child-related risk factors, which include characteristics that arise from the child, such as child age, sex, nutrition status, comorbid conditions, etc. The second level of risk factors is interpersonal/family or caregiver, which is related to maternal and paternal risk exposures. The third level of risk factors is community-related risk factors related to neighbourhoods, schools, peers, early childhood education and care centres. The final fourth level of risk factors are societal risk factors, including policy, program, system, culture and climatic risk factors.”

Journal Requirements

Response: We have revised the manuscript to meet the PLOS ONE Journal requirements. We have also added a data availability section in the revised version of the manuscript (Page 10, Lines 287 & 288). 

“Data availability. 

All relevant data are included in the manuscript.”

Response to the reviewer's comment 

Review #1 Comments to the Author

Comment 1. This is a fair protocol but some details are missing. Adding that clarity now will make the review process easier. There are many studies on this topic so a really comprehensive systematic review, done well, is a helpful contribution to the field. However there are similar entries on PROSPERO, see: here: https://www.crd.york.ac.uk/prospero/display_record.php?RecordID=3215

Response: Thank you for your concern and for providing a similar review protocol on the topic. We aim to provide a comprehensive systematic review mapping the available evidence to date. Unfortunately, we couldn’t find the above PROSPERO registered protocol. But, when we explore it now, it was published in 2012 (i.e., 12 years ago), and still, we couldn’t find the actual systematic review. However, no recent systematic review has provided a comprehensive and in-depth assessment of the literature. Our review will also include adverse developmental outcomes namely developmental delays and developmental vulnerability outcomes, which have not been included in the existing systematic review. We also plan to explore all available risk factors for child development outcomes. 

Comment #2. Please ensure your contribution is a novel contribution to the field. Your registered PROTOCOL uses the word 'adversities' rather than 'vulnerabilities'. The terms need to be consistent.

Response: We have now revised the wording to avoid confusion. Hence, considering the outcomes to be included, we have now revised the title to “Perinatal and early life risk factors of adverse early childhood developmental outcomes: Protocol for systematic review and meta-analysis”. 

Comment #3. Introduction

The concept of early childhood development vulnerability needs to be explained early on. Please define this and if it has been derived from other sources please cite accordingly.

Other studies that have looked at the impact of development at school entry and trajectories of development on adolescent health and may be worth a look if you are making a life course argument https://www.mdpi.com/1660-4601/18/21/11613

and https://www.jpeds.com/article/S0022-3476(23)00474-2/fulltext and want to cite the most recent research. 

Classification system for child development is also discussed here: https://www.mdpi.com/1660-4601/18/21/11613

Author response: We then revised the suggestions and shared important materials. We have revised the introduction and cited relevant literature to support our definition of the outcomes which included developmental vulnerability and developmental delays until the age of 8 on page 3 lines 71-87 and presented as follows:

“Adverse Early child developmental outcomes refer to difficulties or delays in various areas of child development during early childhood periods, including challenges or disruptions in various domains of development, such as physical, cognitive, language, motor, social-emotional, and behavioural areas. Several assessment tools are used to assess early childhood development, such as the Early Development Instrument (EDI), a widely used tool for assessing developmental status at school entry. It assesses child development in five domains: physical health and well-being, social competence, emotional maturity, language and cognitive development, communication skills, and general knowledge. Child development can also be assessed using Ages and Stages Questionnaires (ASQ), which cover communication, gross motor, fine motor, problem-solving and personal social skills. Bayley Scales of Infant and Toddler Development (Bayley), Denver Developmental Screening Test (DDST), Peabody Developmental Motor Scales (PDMS), and Social-emotional assessment tools are also among some of the early childhood developmental assessment tools. These tools vary in format, administration, target age groups and developmental area classifications. Understanding the various domains of early childhood developmental outcomes and their risk factors can inform policy and program development at the community and government levels to better support early childhood development.”

Comment #4. Please provide a simple explanation (and citation) for the socioecological model.

Response: We have now provided citations and a simple explanation of how we apply the socioecological model in our systematic review on page 4, lines 97-106, and page 8, lines 219-232.

 “The various perinatal and early childhood risk factors will be summarized using a four-level socioecological model to better understand early childhood development adversities and potential prevention strategies. The socioecological model is a theoretical framework used to understand and analyze the complex interaction between intrapersonal (individual child characteristics), interpersonal (family/caregiver related characteristics), community (school, peers, teachers, daycares, and neighborhoods related characteristics), and societal (climatic, cultural, systems, policy, and program-related characteristics) level factors28 (Fig 1). The socio-ecological model is an important approach to summarize the various risk factors, which leads to recommend a targeted intervention at each level.” 

comment # 5: The existing evidence on what is known in this field is more comprehensive that the lines portrayed: There were some fragmented systematic syntheses of evidence such as child developmental outcomes associated with low birth weight and small for gestational age in Indigenous Children from Australia, Canada and New Zealand.

Please provide a brief but comprehensive summary on what is known about risk factors (state which ones) that affect ECD vulnerability and state which countries. (This is why your definition of ECD vulnerability is needed right from the outset).

Response: Thank you for your insightful suggestion. We have now summarised the existing systematic review on page 4 lines 112 -126 and defined our outcomes (early childhood outcomes and developmental delays) on page 3 lines 71-87( see the response for comment #3):

 “Previous systematic synthesis have been conducted to assess the risk factors of early childhood developmental outcomes. 27 35-38 However, those studies used either single exposure or a single domain of early childhood developmental adversities. Thus, a comprehensive synthesis of the multilevel risk factors towards adverse early childhood developmental outcomes is limited.39 On the Other hand, the current study will summarize risk factors of adverse early childhood developmental outcomes into individual, interpersonal, and community-level risk factors. In addition, we include all the developmental domains, including physical, social, cognitive, language, communication and other domains based on each assessment tool report. Hence, estimating the global burden of adverse early childhood developmental outcomes and identifying the multilevel associated risk factors is imperative for identifying vulnerable populations and implementing targeted health promotion activities at each level.8 Therefore, this systematic review aimed to estimate the global burden of early childhood developmental vulnerabilities and comprehensively map the broader risk factors of early childhood developmental vulnerability using the socioecological model.”

Comment #6. Research Qs - is research question 2 about predictive risk factors?

Response: Yes, it is about predictive risk factors of adverse early childhood developmental outcomes. 

Comment 7. Eligibility: This section needs to be much clearer. The population blurs with the Concept. Suggest: Population (children 0-5), Exposure (risk factors), Outcomes (developmental) and Study types. See here: https://systematicreviewsjournal.biomedcentral.com/articles/10.1186/s13643-021-01694-6

Response: Thank you for your suggestion. As suggested, we have revised the eligibility criteria to make them clearer on pages 5 and 6 lines 138-166. “We used the Population, Exposure, and Exposure Comparison, Outcome and Studies (PECOCS) criteria to select eligible studies for our study in the following format.

1) Population: Children underwent early childhood developmental assessments under 8 years old. The age of 8 years is selected because early childhood is defined as the period from prenatal development to eight years, according to WHO.

2) Exposure: The exposure of this study includes the various perinatal and early life risk factors. Perinatal and early life risk factors include biological, psychosocial, behavioural, and environmental risk factors associated with early childhood developmental outcomes.

3) Comparison: Children with no or low levels of exposure to the risk factors will be used as a comparator for this study. 

4) Outcomes: The primary outcome of this systematic review is adverse early childhood developmental outcomes such as adverse outcomes of physical, cognitive, social, emotional, language and neurological developments assessed through various assessment tools such as Early Development Instrument (EDI), Age and Stages Questionnaire (ASQ), Bayley Scales of Infant and Toddler Development (Bayley), Denver Developmental Screening Test (DDST), Peabody Developmental Motor Scales (PDMS), and Social-emotional assessment tools etc. 

5) Studies: We will include observational studies such as cohort, case-control, nested case-control, and cross-sectional studies reporting the prevalence or incidence of adverse early childhood developmental outcomes and associated risk factors in children. There will be no restriction based on country of origin or geographical location except language (only published in English) from January 2000. The year 2000 was chosen due to the change in technology, social media use, social and cultural contexts, early childhood education, and environmental exposures over time, which makes it important to incorporate more recent and up-to-date knowledge in the field. There have been substantial changes in policies, interventions, and programs over the last two decades, all aimed at addressing early childhood developmental outcomes.”

comment #8: The risk factors proposed here need to be introduced more clearly in the introduction.

Response: In the revised version, we have described the broad range of risk factors in the introduction section on pages 3 and 4 lines 90-97. “Previous studies showed that early childhood developmental outcomes were affected by various perinatal and early life risk exposures such as genetic and biological factors, prenatal conditions such as maternal health, nutrition, exposure to toxic substances, prenatal care during pregnancy, birth complications, child health, parenting behaviours, access to early childhood education and care, socioeconomic disadvantages, and environmental exposure to toxins or pollutants. A systematic review conducted among Indigenous Children from Australia, Canada and New Zealand showed that child developmental outcomes were associated with low birth weight and small for gestational age”

Comment # 9: Data Synthesis: The terms domains and measurement tools are suddenly introduced. Please be consistent in language and terms throughout the document. 

Author response: We have introduced child developmental domains and developmental assessment tools in the introduction section of the revised manuscript (see page 3, lines 71-87) (this part is presented in the response to comment #3).

Comment #10: It states you expect a variety of early child developmental outcome domains; - why has the language changed from ECD vulnerabilities to developmental outcomes? 

Author response: Thank you for your concerns. Given that child developmental outcomes encompass both child developmental vulnerabilities and developmental delays, we have now preferred to use the term child developmental outcomes to facilitate our comprehensive evaluation of both outcomes. Accordingly, we have now revised the title and aim of the study, as well as clearly defined outcomes.

Comment #11: Your search strategy only lists negative developmental outcomes. Again - this must all be clarified in the eligibility section and then all the terms must remain consistent. E.g. - is the outcome vulnerability (sub-optimal development) or any developmental domain?

Author response: Our primary aim is the adverse developmental outcome for early detection and prevention activities. As some studies will likely report risk factors for one or more sub-domains, we plan to collate the risk factors for sub-outcomes. As a result, we have ensured that our search strategy included keywords for developmental domains. 

Comment #12: It appears there is narrative and quantitative synthesis. Please describe both clearly including a section on confidence in cumulative evidence (i.e. for any narrative synthesis - how will you determine the strength of the evidence?)

Author response: We have revised the data synthesis section as follows (see pages 7 & 8 lines 214-240): “A textual and narrative synthesis of the global burden of early childhood developmental outcomes will be conducted in each developmental domain, such as physical, social, emotional, cognitive, language, communicati

---

## [Decision Letter · Decision Letter 1]

23 Jun 2024

PONE-D-23-35644R1Perinatal and early life risk factors of adverse early childhood developmental outcomes: Protocol for systematic review and meta-analysis.PLOS ONE

Dear Dr. Atalell,

Thank you for submitting your manuscript to PLOS ONE. After careful consideration, we feel that it has merit but does not fully meet PLOS ONE’s publication criteria as it currently stands. Therefore, we invite you to submit a revised version of the manuscript that addresses the points raised during the review process.

Please carefully address the additional comments from the reviewers at this round of review.  Please submit your revised manuscript by Aug 05 2024 11:59PM. If you will need more time than this to complete your revisions, please reply to this message or contact the journal office at plosone@plos.org. Please include the following items when submitting your revised manuscript:A rebuttal letter that responds to each point raised by the academic editor and reviewer(s). You should upload this letter as a separate file labeled 'Response to Reviewers'.A marked-up copy of your manuscript that highlights changes made to the original version. You should upload this as a separate file labeled 'Revised Manuscript with Track Changes'.An unmarked version of your revised paper without tracked changes. You should upload this as a separate file labeled 'Manuscript'.

We look forward to receiving your revised manuscript.

Kind regards,

Jennifer Tucker, PhD

Staff Editor

PLOS ONE

Reviewers' comments:

Reviewer's Responses to Questions

**Comments to the Author**

1. Does the manuscript provide a valid rationale for the proposed study, with clearly identified and justified research questions?

Reviewer #2: Yes

Reviewer #3: Partly

2. Is the protocol technically sound and planned in a manner that will lead to a meaningful outcome and allow testing the stated hypotheses?

Reviewer #2: Partly

Reviewer #3: Partly

3. Is the methodology feasible and described in sufficient detail to allow the work to be replicable?

Reviewer #2: Yes

Reviewer #3: No

4. Have the authors described where all data underlying the findings will be made available when the study is complete?

Reviewer #2: Yes

Reviewer #3: Yes

5. Is the manuscript presented in an intelligible fashion and written in standard English?

Reviewer #2: Yes

Reviewer #3: Yes

6. Review Comments to the Author

You may also provide optional suggestions and comments to authors that they might find helpful in planning their study.

Reviewer #2: Thank you for the opportunity to review this interesting paper, which proposes to conduct a broad and comprehensive systematic review of the literature on early childhood developmental vulnerabilities and perinatal and early life risk factors. This review is much needed and will be a useful tool to researchers studying early child development. The methodological strategy overall is robust and well explained. My main concern is regarding the definition and description of early childhood developmental vulnerability and the perinatal and early life risk factors. These are quite broad and potentially overlapping. The authors will find it difficult to extract meaningful information if they don’t better define and delineate these. It might be useful to attempt to extract a few papers as a pilot exercise to see whether it is feasible. The review is ambitious, and unless they narrow down what they are including in terms of their outcome and exposure, I am concerned about feasibility and clarity of the resulting tables.

The Concept section (#2 page 4) talks about a wide range of risk factors, and it is not until the data synthesis section that we get a sense of what these mean. Recommend having this come earlier.

The authors should better define or perhaps consider narrowing down the early risk exposures. I am especially concerned with including behavioural risk factors and then including behavioural problems in the developmental vulnerability assessments. Many of the measures of the EDI for example includes early life behavioural and psychosocial risk factors. Since the authors are including such a wide range of developmental outcomes (5 or 6? “outcomes of physical, cognitive, social, emotional, language and neurological developments”) it might be better to narrow down the exposures of perinatal risk. If leaving all of these in, they should be clearly defined early on beyond this: “Perinatal and early life risk factors include biological, psychosocial, behavioural, and environmental risk factors associated with early childhood developmental outcomes.” However, I am still concerned that the number of factors to consider in all of these would be absolutely massive, from prenatal smoking, prenatal substance use, low birth weight, parental mental illness/offending, socio-economic disadvantage, neighbourhood crime, child maltreatment, ACEs, early behavioural problems etc. All of these examples could potentially fit in those categories. I am struggling to see how this would be feasible in one systematic review.

Reviewer #3: Thank you for the opportunity to review this protocol paper. I note that it has already been revised based on one set of peer review, and I am grateful to have access to those comments and responses in order to see the paper’s journey thus far.

Overall the paper is not very clearly written, and I share concerns of the original reviewers that there is a lack of clarity around the case for this systematic review in terms of its unique contribution, and in terms of the articulation of the theoretical basis. Whilst I can see improvements have been made in response to initial review, I am not sure all points have been adequately addressed and there is still much to do to improve this paper sufficiently for publication. I will make specific points, major and minor, below.

Abstract

Line 31 – I’m not sure about ‘multifacetedly’ as a word. Suggest remove it or express in more simple terms.

Line 41-42 – why is ‘general knowledge’ being termed as a developmental outcome?

Line 48 – should the outcome not be expressed simply as ‘developmental outcome’ rather than ‘adverse early childhood developmental outcomes’?

Lines 35 & 52 – the former refers to ‘the’ socioecological model, the latter to ‘a’ socioecological model. It is not clear from this exactly which model is being applied, and whether a specific approach is being taken.

Line 56 – suggest ‘in the public domain’ rather than ‘from the evidence base’.

Line 57 – suggest ‘developmental outcomes’ would be a better keyword than ‘developmental vulnerability’.

Introduction

I found the introduction generally lacked a coherent structure and would benefit from revision. The decision to go into detail about a small group of measures from line 74-87 is curious and oddly specific. The sentence beginning ‘these tools vary in format…’ at line 83-84 is the important point, and this should be the main driver of this paragraph.

At line 97, the sentence beginning ‘The various perinatal…’ is premature as the model has not yet been described. In relation to the model, I find it odd that Bronfenbrenner has not once been cited, and that the citations used around the model are often not primary. Further, Figure 1 is rudimentary and inadequate to illustrate the strength of this model. There are many examples online where examples of

the type of factor at each layer are given, and this would certainly be a good approach for a paper offering such a comprehensive review.

On line 95 you refer to a review regarding indigenous children – it is not clear why this very specific example was chosen to illustrate your point.

Line 111 – ‘accessing the quality early childhood’ should be ‘accessing quality early childhood’

Line 113 – plural syntheses should be used instead of synthesis.

Line 123 – suggest paragraph break at ‘therefore’ (although would then revise to not begin sentence with a conjunction).

Research questions

I share the concerns of previous reviewers about the clarity and novelty of these. For example, RQ1 refers to a ‘global estimate’ but it is entirely unclear what metric will be used.

Method

Line 133 – PRISMA is missing the I

Line 144 – under Exposure, why has the term Adverse Childhood Experience (ACE) not been used in searches? Also, you need to stipulate the timing of exposure in relation to outcome.

Lines 152-155 – again, very specific examples of potential measures. It’s not clear why.

Line 156 – Studies – you refer to designs used – what will you do to ensure / assess that samples are population representative (as many studies will be clinical populations and so skew any findings)?

Line 161-164 – this is a perfect opportunity to link to the ‘chronosphere’ dimension of Bronfenbrenner’s model.

Line 172-184 – make it clear that this is your core search strategy, and that specific iterations tailored to each database are available in supplementary material.

Line 199 – will you also record the age @ exposure?

Line 205 – you mention two authors will extract data – will this be double entry from all papers, or a shared workload (i.e., 50:50 split)?

Line 209 – citation 48 is to a secondary source, and the link to the primary is broken. Please update.

Line 216-217 – you write ‘a standard definition will be developed for adverse early childhood developmental outcomes’ as part of your synthesis – but surely this will be developed as part of your inclusion criteria?

Line 225 – you write ‘the four-level socioecological model will be formed based on preliminary literature review’ – what exactly do you mean by this?

Line 233 – you write that you will pool the global burden. Again, using what metric?

Line 236 – polled should read pooled.

Regarding the plans for analysis, as statistical analysis is not my specialism I am not in a position to comment on these.

Overall this protocol may eventually be publishable, but I find it difficult to assess this with the lack of specificity in articulation of the research questions and methods.

7. PLOS authors have the option to publish the peer review history of their article (what does this mean?). If published, this will include your full peer review and any attached files.

Reviewer #2: No

Reviewer #3: **Yes: **Lucy Thompson

---

## [Author Response · Author response to Decision Letter 1]

9 Aug 2024

Ref: PONE-D-23-35644R1

Dr Jennifer Tucker,

Editor, PLOS ONE

Title: Perinatal and early life risk factors of adverse early childhood developmental outcomes: Protocol for systematic review using socioecological model.

Authors: Kendalem Asmare Atalell, Gavin Pereira, Bereket Duko, Sylvester Dodzi Nyadanu, Gizachew A Tessema

Subject: Submission of revised manuscript

Dear Editor,

Thank you for the opportunity to revise our manuscript for the second time, “Perinatal and early life risk factors of adverse early childhood developmental outcomes: Protocol for systematic review and meta-analysis.” We are pleased to revise the manuscript and incorporate the comments. Our responses are given in point-by-point responses to each reviewer and the editor’s comments below. We have also attached the revised manuscript with tracked changes and a clean version.

We hope the revised version is suitable for publication now and look forward to hearing from you soon.

Sincerely

Kendalem Asmare Atalell

Corresponding Author

Response to Reviewers' comments

Response: Thank you for your valuable insights. We aim to provide a comprehensive systematic review identifying various perinatal and childhood risk factors for adverse developmental outcomes. 

Response to Reviewer #2 

1. Thank you for the opportunity to review this interesting paper, which proposes to conduct a broad and comprehensive systematic review of the literature on early childhood developmental vulnerabilities and perinatal and early life risk factors. This review is much needed and will be a useful tool to researchers studying early child development. The methodological strategy overall is robust and well explained. My main concern is regarding the definition and description of early childhood developmental vulnerability and the perinatal and early life risk factors. These are quite broad and potentially overlapping. The authors will find it difficult to extract meaningful information if they don’t better define and delineate these. It might be useful to attempt to extract a few papers as a pilot exercise to see whether it is feasible. The review is ambitious, and unless they narrow down what they are including in terms of their outcome and exposure, I am concerned about feasibility and clarity of the resulting tables.

Response: Thank you for your interest in our comprehensive review; we are very grateful for your valuable concerns. The outcome and exposure are now well explained to clarify what we want to do in our review in the revised manuscript, page 5 lines, lines 129-144: “Exposure: The exposures in this study include various perinatal and early childhood risk factors, such as biological, sociodemographic, child health and medical history, childhood nutrition, maternal mental health issues, pregnancy-related maternal health conditions, maternal nutrition, prenatal exposure to substances and pollutants, environmental pollutants, and climatic factors. 

Outcome: The primary outcome of this systematic review is adverse early childhood developmental outcomes, such as physical, cognitive, language and communication and social-emotional developments,43 44 which is defined as difficulties or delays in various areas of child developmental vulnerability or developmental problems at composite or sub-domain levels.” 

2. The Concept section (#2 page 4) talks about a wide range of risk factors, and it is not until the data synthesis section that we get a sense of what these mean. Recommend having this come earlier.

Response: We have revised the criteria from PCC to PECOS and the risk factors were introduced in the exposure section in the revised version of the manuscript. (See response #1). 

3. The authors should better define or perhaps consider narrowing down the early risk exposures. I am especially concerned with including behavioural risk factors and then including behavioural problems in the developmental vulnerability assessments. Many of the measures of the EDI for example includes early life behavioural and psychosocial risk factors. Since the authors are including such a wide range of developmental outcomes (5 or 6? “outcomes of physical, cognitive, social, emotional, language and neurological developments”) it might be better to narrow down the exposures of perinatal risk. If leaving all of these in, they should be clearly defined early on beyond this: “Perinatal and early life risk factors include biological, psychosocial, behavioural, and environmental risk factors associated with early childhood developmental outcomes.” However, I am still concerned that the number of factors to consider in all of these would be absolutely massive, from prenatal smoking, prenatal substance use, low birth weight, parental mental illness/offending, socio-economic disadvantage, neighbourhood crime, child maltreatment, ACEs, early behavioural problems etc. All of these examples could potentially fit in those categories. I am struggling to see how this would be feasible in one systematic review.

Response: Thank you again for your valuable comments. We also noted that it is a very wide. However, our aim is to summarize all the factors affecting early childhood developmental outcomes using the socioecological model as a framework to better recommend comprehensive and effective health promotion activities at the individual, family, community and societal levels. Hence, we will map all the risk factors of developmental adversities in early childhood. 

Response to Reviewer #3: 

1. Thank you for the opportunity to review this protocol paper. I note that it has already been revised based on one set of peer review, and I am grateful to have access to those comments and responses in order to see the paper’s journey thus far.

Overall the paper is not very clearly written, and I share concerns of the original reviewers that there is a lack of clarity around the case for this systematic review in terms of its unique contribution, and in terms of the articulation of the theoretical basis. Whilst I can see improvements have been made in response to initial review, I am not sure all points have been adequately addressed and there is still much to do to improve this paper sufficiently for publication. I will make specific points, major and minor, below.

Response: Thank you so much for your thoughtful comments 

 Abstract

2. Line 31 – I’m not sure about ‘multifacetedly’ as a word. Suggest remove it or express in more simple terms.

Response: We have now revised the wording to avoid confusion.

3. Line 41-42 – why is ‘general knowledge’ being termed as a developmental outcome?

Response: We have revised the sentence to make it clear for readers as follows developmental outcomes domains in the revised version of the manuscript. The sentence is revised as follows “We will systematically search for articles on adverse early childhood developmental outcomes, which include physical, cognitive, language and communication and social-emotional development from main databases, such as EMBASE, Medline, Global Health, PsycINFO, CINAHL and Web of Science Core Collection from January 2000 to January 2024.” Page 2 lines 37-41.

4. Line 48 – should the outcome not be expressed simply as ‘developmental outcome’ rather than ‘adverse early childhood developmental outcomes’?

Response: Thank you, our outcome is developmental outcome in early childhood periods (under eight years of age). It can be either adverse outcomes such as delay or vulnerability or optimal outcomes when the child reaches his/her potential for his or her age. However, our aim in this systematic review is to identify the risk factors associated with adverse developmental outcomes, which is why we prefer to say adverse early childhood developmental outcomes. 

5. Lines 35 & 52 – the former refers to ‘the’ socioecological model, the latter to ‘a’ socioecological model. It is not clear from this exactly which model is being applied, and whether a specific approach is being taken.

Response: We have revised it and consistently used the socioecological model. The socio-ecological model is briefly introduced in the introduction as well as in the data synthesis section of the manuscript on page 4 lines 101-108 “The socio-ecological model is a theoretical framework used to understand and analyze the complex interaction between 1) Individual factors: factors arise from the individual child such as genetics, health status, and personal behaviours. 2) Interpersonal factors: maternal, paternal, or household-related influences, including family dynamics and parenting styles. 3) Community factors: school, peers, teachers, daycare environments, and neighborhood-related influences. 4) Societal factors: broader influences include climatic conditions, cultural norms, systems, policies, and programs.”

6. Line 56 – suggest ‘in the public domain’ rather than ‘from the evidence base’.

Response: Thank you, Corrected 

7. Line 57 – suggest ‘developmental outcomes’ would be a better keyword than ‘developmental vulnerability’.

Response: We have changed the keyword from developmental vulnerability to developmental outcome; thank you. 

 Introduction

8. I found the introduction generally lacked a coherent structure and would benefit from revision. The decision to go into detail about a small group of measures from line 74-87 is curious and oddly specific. The sentence beginning ‘these tools vary in format…’ at line 83-84 is the important point, and this should be the main driver of this paragraph.

Response: Thank you for the suggestions; we have made a major change in the introduction section, and we believe the introduction is now improved well (See pages 3 and 4. 

9. At line 97, the sentence beginning ‘The various perinatal…’ is premature as the model has not yet been described. In relation to the model, I find it odd that Bronfenbrenner has not once been cited, and that the citations used around the model are often not primary. Further, Figure 1 is rudimentary and inadequate to illustrate the strength of this model. There are many examples online where examples of the type of factor at each layer are given, and this would certainly be a good approach for a paper offering such a comprehensive review.

Response: We have made changes in the introduction and this section is also removed and updated with more suitable descriptions. Thank you. 

10. On line 95 you refer to a review regarding indigenous children – it is not clear why this very specific example was chosen to illustrate your point.

Response: We have removed this specific example from the revised manuscript since our review is not solely based on indigenous populations. 

11. Line 111 – ‘accessing the quality early childhood’ should be ‘accessing quality early childhood’

Response: Corrected 

12. Line 113 – plural syntheses should be used instead of synthesis.

Response: Corrected 

13. Line 123 – suggest paragraph break at ‘therefore’ (although would then revise to not begin sentence with a conjunction).

Response: Corrected 

 Research questions

14. I share the concerns of previous reviewers about the clarity and novelty of these. For example, RQ1 refers to a ‘global estimate’ but it is entirely unclear what metric will be used.

Response: We have revised the research question as follows: “What are the perinatal and early childhood risk factors for adverse early childhood developmental outcomes?” page 4, lines 115-117. 

 Method

15. Line 133 – PRISMA is missing the I

Response: Corrected 

16. Line 144 – under Exposure, why has the term Adverse Childhood Experience (ACE) not been used in searches? Also, you need to stipulate the timing of exposure in relation to outcome.

Response: Thank you. We have done a preliminary literature review before developing the search term, and adverse childhood experience is a general term to be included in the search term. Regarding the timing of the exposure, we plan to include all the exposure time, including the perinatal and childhood exposure. 

17. Lines 152-155 – again, very specific examples of potential measures. It’s not clear why.

Response: There are many developmental assessment tools; we list a few as examples, which are commonly used in many studies as per our preliminary literature review. 

18. Line 156 – Studies – you refer to designs used – what will you do to ensure/assess that samples are population representative (as many studies will be clinical populations and so skew any findings)?

Response: The data extraction checklist included the study setting (clinical or community), data source, and population, which can identify which study is conducted in a clinical population. We plan to conduct a subgroup analysis based on the available data. 

19. Line 161-164 – this is a perfect opportunity to link to the ‘chronosphere’ dimension of Bronfenbrenner’s model.

Response: Bronfenbrenner’s model is more focused on the developmental context of children and the interplay of nested environmental systems, whereas the socioecological model is broader, focusing on health behaviors and outcomes incorporating a more extensive range of social and policy influences. Hence, we plan to use the socioecological model or framework rather than Bronfenbrenner’s model. 

20. Line 172-184 – make it clear that this is your core search strategy, and that specific iterations tailored to each database are available in supplementary material.

Response: We have revised this section to make it clear that this is the core search strategy on page 6 lines 173-175. 

21. Line 199 – will you also record the age @ exposure?

Response: Initially, we plan to collect age at the exposure; after the preliminary literature review, we excluded it because most of the studies are long cohort studies that start in perinatal to childhood, which makes it difficult to capture age exposure. However, we will record age at developmental assessment. 

22. Line 205 – you mention two authors will extract data – will this be double entry from all papers, or a shared workload (i.e., 50:50 split)?

Response: We have revised as follows: “Given the anticipated large volume of studies in our systematic review, we plan to conduct double data extraction for approximately 20% of the included studies. After ensuring consistency in the extracted data, we will proceed to extract the remaining data, with each author handling 50% of the remaining studies.” Page 7 lines 195-199. 

23. Line 209 – citation 48 is to a secondary source, and the link to the primary is broken. Please update.

Response: We have now updated the citations. 

24. Line 216-217 – you write ‘a standard definition will be developed for adverse early childhood developmental outcomes’ as part of your synthesis – but surely this will be developed as part of your inclusion criteria?

Response: Thank you so much for your insights. This section has also been revised, and the outcomes are already defined in the inclusion criteria. (See response #1). 

25. Line 225 – you write ‘the four-level socioecological model will be formed based on preliminary literature review’ – what exactly do you mean by this?

Response: In the socioecological model, the multilevel factors are customised based on the nature of the problem and the data. After doing a preliminary literature search. We customise the socioecological model to four levels. 1) Individual level: is more child level risk factors 2) Interpersonal risk factors are related to maternal, paternal and household level risk factors 3) Community-level risk factors include organizational risks such as school, preschool, and daycare, as well as community and environmental-related risk factors. 4) Societal-level risk factors are risk factors related to national programs, systems and policy-related risk factors for child developmental adversities. This is elaborated in the revised manuscript on page 8, lines 218-226. 

26. Line 233 – you write that you will pool the global burden. Again, using what metric?

Response: Thank you so much. We dropped these research questions based on our preliminary literature search result, we couldn’t be able to get data for the global estimate. 

27. Line 236 – polled should read pooled.

Response: This section has already been removed. 

28. Regard

---

## [Decision Letter · Decision Letter 2]

20 Aug 2024

PONE-D-23-35644R2Perinatal and early life risk factors of adverse early childhood developmental outcomes: Protocol for systematic review using socioecological model.PLOS ONE Dear Dr. Atalell,

Thank you for submitting your manuscript to PLOS ONE. After careful consideration, we feel that it has merit but does not fully meet PLOS ONE’s publication criteria as it currently stands. Therefore, we invite you to submit a revised version of the manuscript that addresses the points raised during the review process.

We look forward to receiving your revised manuscript.

Kind regards,

Tegene Atamenta Kitaw *(MSc, MPH-Epidemiology)*

Academic Editor

PLOS ONE

Journal Requirements:

Reviewers' comments:

Reviewer's Responses to Questions

**Comments to the Author**

1. Does the manuscript provide a valid rationale for the proposed study, with clearly identified and justified research questions?

Reviewer #2: Yes

Reviewer #4: Yes

2. Is the protocol technically sound and planned in a manner that will lead to a meaningful outcome and allow testing the stated hypotheses?

Reviewer #2: Yes

Reviewer #4: Yes

3. Is the methodology feasible and described in sufficient detail to allow the work to be replicable?

Reviewer #2: Yes

Reviewer #4: No

4. Have the authors described where all data underlying the findings will be made available when the study is complete?

Reviewer #2: Yes

Reviewer #4: Yes

5. Is the manuscript presented in an intelligible fashion and written in standard English?

Reviewer #2: Yes

Reviewer #4: Yes

6. Review Comments to the Author

You may also provide optional suggestions and comments to authors that they might find helpful in planning their study.

Reviewer #2: The manuscript is much improved following the revisions. Two final minor points to consider:

Is there an age or timing cut-off for the early life risk factors? Can these be measured at the same time as the outcomes?

I am still struggling to understand how the authors will delineate individual risk factors such as ‘personal behaviours’ p. 4 line 104 from the social-emotional development outcomes. These will be the same measures in many cases. Unless this is no longer one of the risk factors being examined, but rather just part of the theoretical model? This may become an issue at the time of data extraction.

Good luck with this ambitious project!

Reviewer #4: Overall, the protocol is well written. I have a few minor suggestions that I believe could improve the manuscript:

1. In the comparison group, the definition of "low level of exposure to the risk factors" should be clarified.

2. In the JBI quality assessment, it would be better if the author predefined what constitutes a high score for considering a specific study as high quality. Additionally, the author should specify whether extremely low-quality studies will be excluded, and if so, what cutoff point is used to exclude them. Some literature suggests using >75% as high quality and <50% as low quality to exclude studies.

3. There is inconsistency throughout the manuscript regarding whether this review includes a meta-analysis. In some sections, the author refers to both a systematic review and a meta-analysis. If a meta-analysis is included, the methods of synthesis need to be better planned.

4. In the Data Synthesis section, lines 213-218, the statement "The Socioecological model provides..." is irrelevant to this section, as it has already been described in the Introduction.

5. If possible, I suggest the author consider approaches to evaluate the certainty of the evidence that will be generated from this systematic review (e.g., the GRADE approach).

6. In the Discussion section, it would be better if the author discussed the anticipated limitations of this study protocol (e.g., any limitations of the Socioecological model) rather than repeating what was stated in the Introduction.

7. PLOS authors have the option to publish the peer review history of their article (what does this mean?). If published, this will include your full peer review and any attached files.

Reviewer #2: No

Reviewer #4: **Yes: **Ribka Nigatu Haile

---

## [Author Response · Author response to Decision Letter 2]

27 Aug 2024

Response to reviewers’ comments 

PONE-D-23-35644R2

Perinatal and early life risk factors of adverse early childhood developmental outcomes: Protocol for systematic review using socioecological model.

Dear Editor, 

Thank you again for the opportunity to revise our manuscript, “Perinatal and early childhood risk factors of adverse early childhood developmental outcomes: Protocol for systematic review using socioecological model.” We are pleased again to revise the manuscript, which makes improvements. Our responses are given in point-by-point responses to each reviewer and the editor’s comments below. We have also attached the revised manuscript with tracked changes and a clean version.

We hope the revised version is suitable for publication now and look forward to hearing from you soon.

Sincerely

Kendalem Asmare Atalell

Corresponding Author

Response to reviewers’ comment 

Journal Requirements

Please review your reference list to ensure that it is complete and correct. If you have cited papers that have been retracted, please include the rationale for doing so in the manuscript text or remove these references and replace them with relevant current references. Any changes to the reference list should be mentioned in the rebuttal letter that accompanies your revised manuscript. If you need to cite a retracted article, indicate the article’s retracted status in the References list and also include a citation and full reference for the retraction notice.

Response: Thank you so much. We have ensured that all references are correct and complete, and we didn’t cite any retracted articles in our manuscript. 

Reviewer #2 comments: The manuscript is much improved following the revisions. Two final minor points to consider:

Response: Thank you for your positive feedback and for acknowledging the improvements in the manuscript. We appreciate your careful review and will address the comments below. 

Is there an age or timing cut-off for the early life risk factors? Can these be measured at the same time as the outcomes?

Response: In our study, we define early childhood as up to 8 years of age, following the WHO guidelines. We have included cohort, case-control, and cross-sectional studies in our systematic review. Therefore, early childhood risk factors are considered as risk factors reported for children less than 8 years, which corresponds to the period of outcome assessment. In cross-sectional studies, it is possible that risk factors and outcomes are measured at the same time. In cohort and case-control studies, however, risk factors are typically measured at baseline and/or during follow-up, allowing for a temporal separation between risk factors and outcomes. Therefore, we will be acknowledging this inherent limitation in our study.

I am still struggling to understand how the authors will delineate individual risk factors such as ‘personal behaviours’ p. 4 line 104 from the social-emotional development outcomes. These will be the same measures in many cases. Unless this is no longer one of the risk factors being examined, but rather just part of the theoretical model? This may become an issue at the time of data extraction.

Response: Thank you for your insightful feedback. We acknowledge the importance of clearly delineating ‘personal behaviours’ from ‘social-emotional’ developmental outcomes. We have revised the personal behaviour-related factors into lifestyle and behavioural factors, which include the presence of childhood books and storytelling, toys, childhood punishment, mineral and vitamin supplementation, excessive screen time, physical activity, sleeping patterns, etc. While social-emotional development is one of the domains of developmental outcome reported in most developmental measurement tools as we have observed in the preliminary literature review on the topic. 

Response to Reviewer #4 comments: Overall, the protocol is well written. I have a few minor suggestions that I believe could improve the manuscript:

Response: Thank you for the positive feedback on our protocol. We appreciate your suggestions for improvement.

1. In the comparison group, the definition of "low level of exposure to the risk factors" should be clarified.

Response: In our study, "the comparators are children with no or low level of exposure to the risk factors" refers to individuals or groups who have minimal exposure to the identified risk factors compared to the general population; this will be determined based on specific thresholds or criteria established in the included studies. Some studies might have exposure levels like low, medium and high; in that case, we will use low levels of exposure as a comparator, particularly in environmental exposures. 

2. In the JBI quality assessment, it would be better if the author predefined what constitutes a high score for considering a specific study as high quality. Additionally, the author should specify whether extremely low-quality studies will be excluded, and if so, what cutoff point is used to exclude them. Some literature suggests using >75% as high quality and <50% as low quality to exclude studies.

Response: There are still arguments to include or exclude the studies with low-quality scores. Hence, we plan to do a sensitivity analysis if the data will allow us, or we will conduct a descriptive analysis based on the study JBI quality score rather than excluding studies with low quality. We have delineated the score into high, moderate and low for >75, between 50-75 and <50, respectively. 

3. There is inconsistency throughout the manuscript regarding whether this review includes a meta-analysis. In some sections, the author refers to both a systematic review and a meta-analysis. If a meta-analysis is included, the methods of synthesis need to be better planned.

Response: Previously, we thought to conduct a meta-analysis, but now after we had a preliminary literature review, we thought the data would not allow us to conduct a meta-analysis; rather, informed by the socioecological model, we will undertake narrative synthesis to comprehensively summarise the risk factors. We have revised the manuscript, and we have now removed the term meta-analysis. 

4. In the Data Synthesis section, lines 213-218, the statement "The Socioecological model provides..." is irrelevant to this section, as it has already been described in the Introduction.

Response: We have now removed the redundant statement about the Socioecological model from the Data Synthesis section.

5. If possible, I suggest the author consider approaches to evaluate the certainty of the evidence that will be generated from this systematic review (e.g., the GRADE approach).

Response: Thank you for your valuable suggestion. We agree that evaluating the certainty of the evidence generated from our systematic review is crucial for providing robust and transparent conclusions. Hence, we plan to use the GRADE (Grading of Recommendations, Assessment, Development, and Evaluations) approach in our analysis to assess the quality of evidence across the included studies. 

6. In the Discussion section, it would be better if the author discussed the anticipated limitations of this study protocol (e.g., any limitations of the Socioecological model) rather than repeating what was stated in the Introduction.

Response: We have revised to include the anticipated limitations of our systematic review in the discussion sections as follows: “One of the anticipated limitations of our systematic review is the heterogeneity in early childhood developmental assessment tools, which could lead to variability in how developmental outcomes and their domains are reported. This diversity might introduce challenges in synthesising the data and drawing consistent conclusions across studies. To address this, we will employ the socioecological model, which provides a systematic group of risk factors, enhancing a multilevel targeting approach.

---

## [Editor Report · Decision Letter 3]

19 Sep 2024

Perinatal and early life risk factors of adverse early childhood developmental outcomes: Protocol for systematic review using socioecological model.

PONE-D-23-35644R3

Dear Dr. Atalell,

We’re pleased to inform you that your manuscript has been judged scientifically suitable for publication and will be formally accepted for publication once it meets all outstanding technical requirements.

Kind regards,

Tegene Atamenta Kitaw, MSc,MPH

Academic Editor

PLOS ONE

---

## [Editor Report · Acceptance letter]

8 Oct 2024

PONE-D-23-35644R3 

PLOS ONE

Dear Dr. Atalell, 

I'm pleased to inform you that your manuscript has been deemed suitable for publication in PLOS ONE. Congratulations! Your manuscript is now being handed over to our production team.

Kind regards, 

on behalf of

Dr. Tegene Atamenta Kitaw 

Academic Editor

PLOS ONE